# Risk of Dementia in Patients with Leptospirosis: A Nationwide Cohort Analysis

**DOI:** 10.3390/ijerph16173168

**Published:** 2019-08-30

**Authors:** Chun-Hsiang Chiu, Po-Chung Chen, Ying-Chuan Wang, Cheng-Li Lin, Feng-You Lee, Chia-Chang Wu, Kuang-Hsi Chang

**Affiliations:** 1Division of Infectious Diseases and Tropical Medicine, Department of Internal Medicine, Tri-Service General Hospital, National Defense Medical Center Taipei, Taipei 11490, Taiwan; 2Institute of Clinical Medicine, School of Medicine, National Yang-Ming University, Taipei 11221, Taiwan; 3Division of Family Medicine, Department of Community Medicine, Taoyuan Armed Forces General Hospital, Taoyuan 32549, Taiwan; 4Department of Family Medicine, Tri-Service General Hospital, National Defense Medical Center, Taipei 11490, Taiwan; 5Management Office for Health Data, China Medical University Hospital, Taichung 40402, Taiwan; 6College of Medicine, China Medical University, Taichung 40402, Taiwan; 7Department of Emergency Medicine, Taichung Tzu Chi Hospital, Taichung 42743, Taiwan; 8Department of Urology, School of Medicine, College of Medicine, Taipei Medical University, Taipei 11042, Taiwan; 9Department of Medical Research, Tungs’ Taichung Metroharbor Hospital, Taichung 43503, Taiwan; 10Institute of Biomedical Sciences, China Medical University, Taichung 40402, Taiwan; 11General Education Center, Jen-Teh Junior College of Medicine, Nursing and Management, Miaoli 35664, Taiwan

**Keywords:** leptospirosis, dementia, risk factor, vascular hypothesis

## Abstract

Background: Studies have linked some bacterial infections with an increased likelihood for development of dementia. However, there is a paucity of data on the relationship between dementia and leptospirosis. In view of this, we conducted a retrospective cohort study to determine whether leptospirosis is a risk factor for dementia. Methods: Data were collected from the Taiwan National Health Insurance Research Databases (2000–2010) to investigate the incidence of and risk factors for dementia in patients with leptospirosis. Patients with leptospirosis who did not have a history of dementia were enrolled in the study. For each leptospirosis patient, four controls were randomly selected after frequency matching of age, sex, and index date. Cox proportional hazard regression models were used for the analyses of dementia risk. Results: A greater risk of dementia was observed in the leptospirosis cohort than in the non-leptospirosis cohort both in patients without any comorbidity (adjusted HR (aHR) = 1.23, 95% CI = 1.06–1.43) and with a comorbidity (aHR = 2.06, 95% CI = 1.7–2.5). Compared with the non-leptospirosis cohort without these comorbidities, the leptospirosis cohort with ≥2 comorbidities exhibited a significantly increased risk of dementia (aHR = 6.11, 95% CI = 3.15–11.9), followed by those with any one comorbidity (adjusted HR = 3.62, 95% CI = 1.76–7.46). Conclusions: Patients with leptospirosis were at a 1.89-fold greater risk of subsequent dementia, but potential genetic susceptibility bias in the study group is a major confound.

## 1. Background

Leptospirosis, one of the most common zoonotic infections worldwide [1], is caused by a strain of the *Leptospira* bacterium. Leptospires are Gram-negative spirochetes that comprise 24 serogroups and more than 250 serovars. Although they are usually encountered in the tropics and developing countries [2], leptospirosis occurs worldwide through occupational hazards, recreational hazards, and waste hazards [3]. The clinical course of leptospirosis varies. Most cases are mild and self-limited or subclinical, whereas some cases result in Weil’s disease (the triad of jaundice, acute renal failure, and bleeding), hemolytic crisis, and multiorgan failure [4].

Dementia is a neurodegenerative disorder characterized by progressive deterioration in cognitive function. The major underlying pathologies are Alzheimer’s disease (AD), vascular dementia, frontotemporal lobar degeneration, and Lewy body dementia. AD is the most common form of dementia, occurring in 11% of the elderly population aged ≥65 years and 32% of the elderly population aged ≥85 years in the United States [5]. Vascular dementia is the second most common type of dementia, affecting 10%–50% of all dementia cases [6], with a prevalence of nearly 14.5% among US adults aged ≥65 years [7]. There is growing evidence that endothelial dysfunction and oxidative stress play a role in the pathogenesis of both forms of dementia [8,9,10]. Due to oxidative stress and endothelial dysfunction being crucial events in the pathogenesis of leptospirosis, leptospirosis may be a risk factor for diseases related to endothelial dysfunction, such as dementia. A previous study demonstrated the relationship between leptospirosis and acute coronary artery disease [11]. However, data on the relationship between dementia and infections are scant. Thus, we conducted a retrospective cohort study to determine whether leptospirosis is one of the potential risk factors for dementia.

## 2. Materials and Methods

### 2.1. Data Sources

The Taiwan National Health Insurance (NHI) program, launched in March 1995, contains the healthcare data of more than 99% of Taiwan’s population of 23 million [12]. The National Health Insurance Research Databases (NHIRD) are administrative databases containing the claims records from the NHI program [13]. The details of the NHIRD have been previously documented [14,15]. The NHIRD consists of de-identified secondary data released for research purposes. We used the identification of residents to link two data files: a subset of the NHIRD that included inpatient claims, and a registry of beneficiaries. All data files were linked to encrypted identifications to secure the privacy of the insurant. The Ethics Review Board of the China Medical University and Hospital in Taiwan approved this study (CMUH-104-REC2-115). The International Classification of Disease, Ninth Revision, Clinical Modification (ICD-9-CM) was used for defining the disease diagnosis codes.

### 2.2. Study Participants

Patients newly diagnosed with leptospirosis (ICD-9-CM code 100) from 2000 to 2010 were identified as the leptospirosis cohort. Patients in Taiwan suspected to have leptospirosis on the basis of possible exposure history and typical symptoms were studied by two methods: culture isolation and serological diagnosis [16]/ For culture of *leptospira*, blood was taken during the first 10 days of disease and urine was taken 10 days later since the onset of disease. For serological diagnosis, paired acute and convalescent sera were performed with a microscopic agglutination test (MAT). MAT is performed by incubating patient serum with various serovars of leptospires. MAT titer is obtained by testing various serum dilutions with the positive serovar. The serovar that reacts with patient serum is suggested to be the infecting serovar. A positive laboratory diagnosis of leptospirosis required one of the following two criteria: 1) culture isolation or 2) serological diagnosis by a four-fold rise in titer between the acute phase and the convalescent phase and a titer ≥ 1:400 in a single serum. Laboratory studies were all performed at Centers of Disease Control of Taiwan. The initial date of diagnosis of leptospirosis was set as the index date for estimating the durations of follow-ups. Patients with a history of dementia (ICD-9-CM codes 290, 294.1, and 331.0) before the index date, or missing information on age or sex, were excluded. A cohort of individuals with no history of leptospirosis was randomly selected from a population of insured people, at a case-control ratio of 1:4. Both groups were then frequency-matched according to age (every 5-y span), sex, and index year using the same exclusion criteria. All participants were followed until the day of diagnosis of dementia, censoring for loss to follow-up, death, termination of insurance, or December 31, 2011. Preexisting comorbidities included diabetes mellitus (ICD-9-CM code 250), hypertension (ICD-9-CM codes 401–405), head injury (ICD-9-CM codes 310.2, 800, 801, 803, 804, 850, 851, 853, and 854), depression (ICD-9-CM codes 296.2, 296.3, 296.82, 300.4, and 311), stroke (ICD-9-CM codes 430–438), and chronic obstructive pulmonary disease (COPD; ICD-9-CM codes 490–492, 494, and 496).

### 2.3. Statistical Analysis

Distributions of sex, age (≤49 y, 50–64 y, and ≥65 y), and comorbidities were compared between the leptospirosis cohort and non-leptospirosis cohort and examined using the chi-square test. The mean ages of both cohorts were measured and compared using the Student’s t-test. The Kaplan–Meier method was used to assess the cumulative incidence of dementia between the leptospirosis cohort and the non-leptospirosis cohort, and their differences were estimated using a log-rank test.

We performed stratified and multivariate analysis to eliminate confounding effects [17]. The overall incidence densities and sex-, age-, and comorbidity-specific incidence densities of dementia (per 1000 person-years) were calculated for each cohort. Univariable and multivariable Cox proportional hazard regressions models were used for measuring the overall risk and sex-, age-, and comorbidity-specific risk of dementia associated with leptospirosis. Hazard ratios (HRs) and 95% confidence interval (CIs) were estimated using the Cox model. Further data analysis was conducted in patients with leptospirosis and a comorbidity, to evaluate the synergistic effect of both on the risk for dementia. All analyses were conducted using SAS Version 9.4 (SAS Institute, Cary, NC, USA). A *p*-value <0.05 was considered statistically significant.

## 3. Results

The leptospirosis cohort comprised 3766 patients with leptospirosis, and the non-leptospirosis cohort comprised 15,064 sex- and age-matched controls, both with similar distributions in sex and age. The study population exhibited a greater number of males (68.5%), while 45.2% of the participants were aged <49 years. The mean ages of patients in the leptospirosis and non-leptospirosis cohorts were 52.4 ± 16.3 years and 52.0 ± 16.5 years, respectively. The leptospirosis cohort exhibited a significantly higher prevalence of diabetes mellitus, hypertension, head injury, depression, stroke, and COPD than the non-leptospirosis cohort did (all *p* < 0.001), as shown in Table 1. The mean duration of follow-up was 3.71 ± 2.59 years and 4.28 ± 2.49 years in the leptospirosis cohort and non-leptospirosis cohort, respectively (Table 1).

During the 13,989 and 64,400 person-years of follow-up, the overall incidence density of dementia was significantly higher in the leptospirosis cohort than in the non-leptospirosis cohort (1.93 vs. 1.01 per 1000 person-years) with a crude HR of 1.91 (95% CI = 1.71–2.13, Table 2).

After adjustment for age, sex, and the comorbidity of diabetes mellitus, the leptospirosis cohort exhibited a 1.89-fold increased risk of dementia (95% CI = 1.72–2.08). In both cohorts, the incidence of dementia was higher in females than in males, higher in patients with comorbidities than in those without any comorbidity, and increased with aging.

The sex-specific relative risk of dementia was higher in the leptospirosis cohort than in the non-leptospirosis cohort in both females (adjusted HR (aHR) = 2.69, 95% CI = 2.29–3.15) and males (aHR = 1.54, 95% CI = 1.36–1.74). Across all age groups, the leptospirosis cohort was significantly associated with a greater risk of dementia than the non-leptospirosis cohort (aHR = 1.53, 95% CI = 1.34–1.74 in participants aged ≤65 years; aHR = 1.73, 95% CI = 1.42–2.10 in participants aged >65 years, respectively). The comorbidity-specific aHRs of dementia in the leptospirosis cohort compared with the non-leptospirosis cohort were 1.23 (95% CI = 1.06–1.43) and 2.06 (95% CI = 1.70–2.50) in patients without and with comorbidities, respectively.

The results of Kaplan–Meier analysis revealed that the leptospirosis cohort exhibited a higher cumulative incidence of dementia than the non-leptospirosis cohort did (log-rank test, *p* < 0.001, Figure 1).

## 4. Discussion

In Table 1, we observed a greater percentage of males (68.5%) than females (31.5%). This difference in sex-specific incidence is compatible with the finding of a previous study in Taiwan [18], and is explained by male subjects having a higher exposure to leptospirosis risk factors associated with male-dominated occupations, such as livestock farming, fishing, and butchering.

Studies have linked some infectious diseases with an increased likelihood for the development of dementia. However, studies on the relationship between dementia and leptospirosis are few. To our knowledge, this is the first study to demonstrate that patients with leptospirosis exhibit a 1.89-fold greater risk of subsequent dementia than the general population.

The clinical course of leptospirosis is variable. Some cases are mild and self-limited or subclinical, while some are severe and potentially fatal. It is hard to imagine the association between mild leptospirosis and the following chronic disease. However, in leptospirosis patients with severe symptoms who require hospitalization, *leptospires* may cause cytokine storm, endothelial damage, vascular hypo-perfusion, and subsequent organ failure [19]. We highlighted the role of endothelial damage and vascular hypo-perfusion in subsequent chronic disease such as dementia. Thus, we chose the leptospirosis patients with severe symptoms who require hospitalization for our study.

Dementia is a term that describes various progressive, neurodegenerative disorders characterized by a decline in cognitive function. AD is the most common cause of dementia, followed by vascular dementia [5] Unlike vascular dementia, which in most cases, occurs as a result of blood vessel blockage or intracerebral hemorrhage, the exact etiology of AD remains unclear. The major hypothesis on the etiology of AD is the amyloid cascade hypothesis [20]. However, the theory of senile plaque accumulation being the primary cause of neurodegeneration in AD has been questioned in previous studies [21,22]. Whether the formations of senile plaques and neurofibrillary tangles (NFTs) are the cause of AD, or simply occur as a reactive response, remains unknown. Increasing evidence suggests that NFTs may be induced by, and reflect, infection of the central nervous system [23]. Some studies have reported that several people with normal cognitive function also had abundant senile plaques or NFTs in their brains [24,25]. Moreover, studies on disease-modifying strategies targeting the Aβ pathway have shown a lack of efficacy [26,27,28]. Therefore, the overexpression of APP and amyloid deposition might be the result of an acute-phase response to neuronal damage. In recent years, the Aβ oligomer hypothesis, which states that oligomers are the initiating pathologic agents in Alzheimer’s disease, has all but supplanted the amyloid cascade hypothesis, which suggested that fibers were the key etiologic agents in Alzheimer’s disease [29,30,31,32]. There is also a growing body of evidence indicating that vascular dysfunction plays a pivotal role in the development of AD. Epidemiological studies, including the Honolulu-Asia Study, Rotterdam Study and Kungsholmen Project have showed that several vascular risk factors, such as hypertension, atherosclerosis, hyperlipidemia, and stroke, are associated with AD-type dementia [33,34,35]. The significant association between vascular diseases and AD suggests that dementia associated with AD may be caused by vascular mechanisms.

Pathogenesis of leptospirosis also involves endothelial damage and vascular hypo-perfusion [36]. Leptospires frequently enter the body through abraded skin and are distributed into the bloodstream without the initial inflammatory response, followed by an immune phase characterized by antibody production [36]. At this stage, endothelial damage, vasculitis, and inflammatory cell infiltration may occur due to oxidative stress in any tissue [4]. Our study revealed that the incidence of dementia in patients with and without comorbidities (diabetes mellitus, hypertension, head injury, depression, stroke, and COPD) was higher in the leptospirosis cohort than in the non-leptospirosis cohort. The aHRs of the leptospirosis cohort compared with the non-leptospirosis cohort were 1.23 and 2.06 in patients without and with comorbidities, respectively. This finding revealed that leptospirosis might share at least some risk factors involved in the pathogenesis of dementia, such as endothelium dysfunction. As we know, leptospirosis may be one of the most under-recognized, neglected, but easily treatable disease in the world [37,38,39]. Persons who engage in occupational or recreational behavior that brings them into contact with *Leptospira*-infested habitats, may suffer from leptospirosis frequently with or without clear diagnosis [40]. This kind of chronic exposure to leptospirosis may be the pathogenesis of developing chronic endothelial damage, inflammatory processes, and the subsequent dementia.

The potential linkages between other pathogens and AD had been addressed in many earlier studies. Hepatitis C viral infection, Helicobacter pylori infection, Herpes simplex type 1 infection, cytomegalovirus infection, neurospirochetosis, chronic osteomyelitis, or even sepsis, were associated with an increased risk of dementia [41,42,43,44,45,46,47,48]. These studies highlighted the association between chronic or repeated inflammatory processes in brain and dementia.

In support of a genetic bias, in our patient group, comorbidities such as diabetes, COPD, and depression, conditions that have all been postulated to be associated with infection [49,50,51], were all over-represented in the Leptospirosis group: relative prevalences were diabetes, 2.75; COPD, 2.1; and depression, 3.21 (Table 1), relative risks that were comparable to or even exceed the differential risk reported here for Leptospirosis (1.89).

Otherwise, the risk of dementia was observed to have increased in leptospirosis patients with the comorbidities in our study, suggesting that these comorbidities such as diabetes, hypertension, head injury, depression, stroke, and COPD may contribute at least partially to the pathogenesis of dementia, such as endothelium damage or vascular hypo-perfusion.

We enrolled a large sample of participants because the NHI program is universal and mandatory in Taiwan. The NHI beneficiaries have been assigned unique personal identification numbers that enable researchers to trace the details of participants throughout the follow-up period of the study.

The present study had several limitations that must be considered. First, the diagnosis of dementia and leptospirosis were obtained based on the ICD-9-CM codes instead of validated structural diagnostic instruments. This data may be less accurate in the enrolled subjects. Second, the history of the leptospirosis cohort showed a higher prevalence of pre-existing comorbidities. Although we adjusted the HR by comorbidities, the history of more pre-existing comorbidities may have caused detection bias because of frequent clinical visits and examinations. Third, the diagnosis of leptospirosis was determined using ICD-9-CM codes from the NHIRD. However, leptospirosis infections that presented as mild symptoms may have been coded incorrectly and were not enrolled. Fourth, the leptospirosis patients that enrolled in our study were hospitalized patients who possibly had relatively severe symptoms. Therefore, the increased risk of dementia may only be observed in leptospirosis patients with severe symptoms who require hospitalization. Fifth, information on education level, occupational history, socioeconomic status, individual behavior (e.g., smoking habits and exercise habits), drug history, and family history are not recorded in the NHIRD. These factors may be crucial in the development of dementia. Sixth, because of potential genetic bias governing disease susceptibility in the patient versus control groups, our study falls short of providing evidence for a causal role of *Leptospira* infection in later dementia development; further research will be necessary to unravel the associations between specific infectious agents, infection susceptibility, and dementia.

## 5. Conclusions

In conclusion, our study revealed that patients with leptospirosis are at a 1.89-fold greater risk of subsequent dementia than the general population. It is important to ensure people with occupational or recreational hazards are aware of the risk factors, symptoms, and possible complications of leptospirosis. Future studies are needed to confirm this epiphenomenon. 

## Figures and Tables

**Figure 1 ijerph-16-03168-f001:**
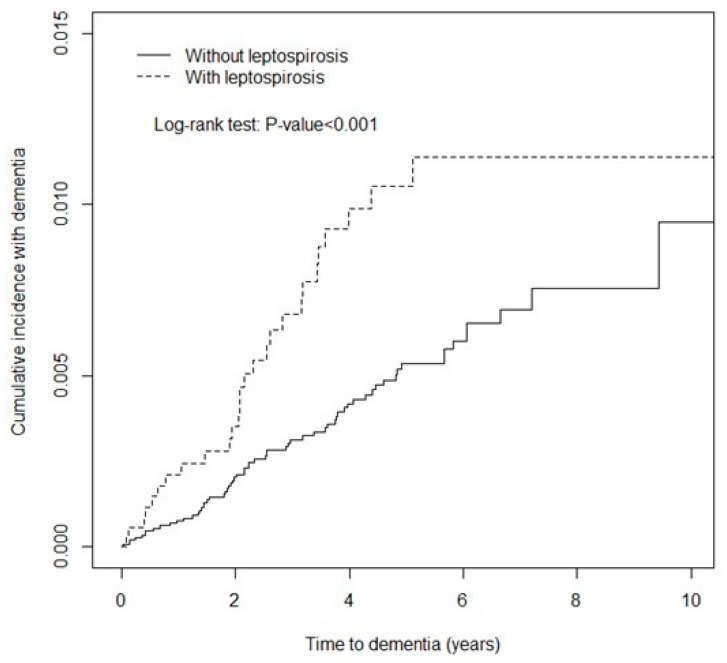
Comparison of the cumulative incidence of dementia in the leptospirosis cohort and non-leptospirosis c.

**Table 1 ijerph-16-03168-t001:** Demographic characteristics and comorbidities in patients with and without leptospirosis.

Variable		Leptospirosis (n = 3766)	Control (n = 15,064)	*p*-value
Sex				
Female		1185 (31.5)	4740 (31.5)	0.99
Male		2581 (68.5)	10,324 (68.5)	
Age, mean	Mean (SD)	52.4 (16.3)	52.0 (16.5)	0.20
Age groups (years)	≤49	1127 (45.2)	6804 (45.2)	0.99
	50–64	1127 (29.9)	4508 (29.9)	
	≥65	938 (24.9)	3752 (24.9)	
Comorbidity				
Diabetes		570 (15.1)	827 (5.5)	<0.001
Hypertension		824 (21.9)	1444 (9.6)	<0.001
Head injury		236 (6.3)	503 (3.3)	<0.001
Depression		70 (1.9)	88 (0.6)	<0.001
Stroke		301 (8.0)	596 (4.0)	<0.001
COPD		206 (5.5)	393 (2.6)	<0.001

COPD: Chronic Obstructive Pulmonary Disease.

**Table 2 ijerph-16-03168-t002:** Comparison of incidence and hazard ratios of dementia stratified by sex, age, and comorbidity between the leptospirosis and non-leptospirosis patients.

Variable	Leptospirosis			Control			Crude HR* (95% CI)	Adjusted HR† (95% CI)
	No. of dementia	PY	Rate#	No. of dementia	PY	Rate#		
All	27	13,989	1.93	65	64,400	1.01	1.91 (1.71, 2.13)	1.89 (1.72, 2.08)
Sex								
Female	12	4447	2.70	22	20,138	1.09	2.47 (2.06, 2.96)	2.69 (2.29, 3.15)
Male	15	9542	1.57	43	44,262	0.97	1.62 (1.41, 1.86)	1.54 (1.36, 1.74)
Age (years)								
≤65	3	11,074	0.27	5	48,961	0.10	2.65 (2.32, 3.03)	1.53 (1.34, 1.73)
>65	24	2915	8.23	60	15,439	3.89	2.12 (1.74, 2.58)	1.73 (1.42, 2.10)
Comorbidity‡								
No	4	9145	0.44	29	55,001	0.53	0.84 (0.29, 2.38)	1.23 (1.06, 1.43)
Yes	23	4844	4.75	36	9399	3.83	1.23 (0.73, 2.07)	2.06 (1.70, 2.50)

No. of dementia: number of patients with dementia; PY: person-years; Rate#: incidence rate, per 1000 person-years; Crude HR*: crude hazard ratio; Adjusted HR†: multivariable analysis including age, sex, and comorbidities of diabetes, hypertension, head injury, depression, stroke, and COPD Comorbidity‡: Patients with any one of the comorbidities (diabetes, hypertension, head injury, depression, stroke, and COPD) were classified as the comorbidity group.

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
