# Peer review of "Risk of Dementia in Patients with Leptospirosis: A Nationwide Cohort Analysis"

_ijerph, 2019, doi:10.3390/ijerph16173168_

Round 1

Reviewer 1 Report

ABSTRACT

The abstract is well-organized and the purpose of the study is clear. I would change just one thing. the Authors stated “Studies have linked some viral infections with an increased likelihood for development of dementia”. Leptospirosis is an infection caused by the bacterium Leptospira, then it would be more appropriate to connect bacterial infections (instead of viral infections) to dementia.

KEYWORDS

“amyloid cascade hypothesis” should be removed from the keywords. The Authors mentioned it just once (Line 176) but its “weight” in the present manuscript is very low (I would say “close to zero”).

INTRODUCTION

The introduction can be slightly improved.

Please add at least one reference at the end of Line 57.

The Authors stated “There is growing evidence that endothelial dysfunction and oxidative stress play a role in the pathogenesis of both forms of dementia”. Since oxidative stress also plays a significant role in leptospirosis (PMID: 24493675), this “bridge” connecting all those diseases should be expanded/highlighted in the Introduction section. Oxidative stress should also be mentioned in the Discussion section within the part “Lines 186-188”.

MATERIALS AND METHODS

The Authors did not add any reference at the end of “Patients in Taiwan suspected to have leptospirosis [...] were studied by two methods: culture isolation and serological diagnosis”. Please add more details.

Same for “Paired acute and convalescent sera were performed with microscopic agglutination test”. Please add more details.

The Authors should explain their choice (or at least add a reference) to use “Univariable and multivariable Cox proportional hazard regression models” to meausre the overall risk and sex-, age-, and comorbidity-specific risk of dementia associated with leptospirosis.

RESULTS

The results are clearly described with the exception of Table 2; it is very confusing for the readers and should be reorganized.

DISCUSSION

The references are totally missing in the first (big) part (Lines 160-173) of the Discussion section.

The Authors stated “The major hypothesis on the etiology of AD is the amyloid cascade hypothesis”. Well, there are many (and very current) publications (review, in vitro, and in vivo studies) giving more relevance to the Aβ oligomer hypothesis (PMID: 29843241), which show how the toxicity/neurodegeneration observed in AD mainly depends on the formation of oligomeric species (PMID: 27804051; PMID: 30497524; PMID: 31293421). This “change of direction” should be emphasized in the Discussion section. Additionally, in certain conditions (e.g. sub-toxic levels or monomeric form) Aβ peptide could be protective for neurons by inducing the release of neurotrophic factors (PMID: 29094448; PMID: 30463298).

The sentence “Whether the formations of senile plaques and NFTs are the causes of AD, or simply occur as a reactive response, remains unknown, although increasing evidence suggests that they may be induced by, and reflect, infection of the central nervous system” should be reformulated. Please define “NFT” before using it as an acronym.

A reference should be added at the end of “Pathogenesis of leptospirosis also involves endothelial damage and vascular hypo-perfusion” and of “Leptospires frequently enter the body through abraded skin and are distributed into the bloodstream without the initial inflammatory response, followed by an immune phase characterized by antibody production”.

I could not find any connection with the Tables or Figures (except for Table 1). The Authors should refer to a specific table or figure (when it is pertinent) during their discussion, it will help the readers to better follow the “speech flow”.

A reference should be added at the end of “This finding revealed that leptospirosis might share at least some risk factors involved in the pathogenesis of depression, such as endothelium dysfunction”.

The meaning of the sentence “This kind of chronic exposure to leptospirosis maybe the pathogenesis of developing chronic endothelial damage, inflammatory processes, and the subsequent dementia” is not clear.

The authors should tone down some of their statements, perhaps by deleting or reformulating “The association of dementia with leptospirosis and C. pneumoniae infection may also suggest that the vascular hypothesis may delineate the real pathogenesis of AD”. Additionally, why do they start talking about C. pneumoniae infection at some point? They have dedicated an entire paragraph to it (too much). It is distracting.

Based on the proposed title “Risk of dementia in patients with leptospirosis: A nationwide cohort analysis”, the Discussion section related to AD pathology is unnecessarily long.

CONCLUSION

The conclusions are, at least in part, supported by Authors’ results. As highlighted by the Authors (I really like that they did this, almost nobody does it), “The present study had several limitations that must be considered”. As previously mentioned, if the Authors tone down some of their statements the present study could represent a good “pilot” for deeper future investigations.

Minors:

1) Leptospira, Chlamydia pneumoniae, etc. should be “Italic”

2) It not clear why some parts of the references are in bold. The Authors should be consistent with the references format.

Author Response

ABSTRACT

The abstract is well-organized and the purpose of the study is clear. I would change just one thing. the Authors stated “Studies have linked some viral infections with an increased likelihood for development of dementia”. Leptospirosis is an infection caused by the bacterium Leptospira, then it would be more appropriate to connect bacterial infections (instead of viral infections) to dementia.

Reply: Thank you for your comments. We’ve revised as suggested. (page 3, line 3)

KEYWORDS

“amyloid cascade hypothesis” should be removed from the keywords. The Authors mentioned it just once (Line 176) but its “weight” in the present manuscript is very low (I would say “close to zero”).

Reply: Thank you for your comments. We’ve deleted it from keywords..

INTRODUCTION

The introduction can be slightly improved.

Please add at least one reference at the end of Line 57.

Reply: Thank you for your comments. We’ve added reference 13. (page 5, line 59)

The Authors stated “There is growing evidence that endothelial dysfunction and oxidative stress play a role in the pathogenesis of both forms of dementia”. Since oxidative stress also plays a significant role in leptospirosis (PMID: 24493675), this “bridge” connecting all those diseases should be expanded/highlighted in the Introduction section.

Reply: Thank you for your comments. We’ve revised as suggested. (page4, line44-48)

Oxidative stress should also be mentioned in the Discussion section within the part “Lines 186-188”.

 Reply: Thank you for your comments. We’ve revised as suggested. (page 10, line199-201)

MATERIALS AND METHODS

The Authors did not add any reference at the end of “Patients in Taiwan suspected to have leptospirosis [...] were studied by two methods: culture isolation and serological diagnosis”. Please add more details.

Reply: Thank you for your comments. We’ve added reference 16. (page5, line73)

Reply: Thank you for your comments. We’ve added more details about culture isolation and serological diagnosis (page 5-6, line73-79)

Same for “Paired acute and convalescent sera were performed with microscopic agglutination test”. Please add more details.

Reply: Thank you for your comments. We’ve added more details about culture isolation and serological diagnosis (page 5-6, line75-79)

The Authors should explain their choice (or at least add a reference) to use “Univariable and multivariable Cox proportional hazard regression models” to meausre the overall risk and sex-, age-, and comorbidity-specific risk of dementia associated with leptospirosis.

 Reply: Thank you for your comments. We’ve added reference 17. (page7, line 106-107)

RESULTS

The results are clearly described with the exception of Table 2; it is very confusing for the readers and should be reorganized.

 Reply: Thank you for your comments. We’ve restructure the paragraph of Results .

DISCUSSION

The references are totally missing in the first (big) part (Lines 160-173) of the Discussion section.

Reply: Thank you for your comments. We’ve added reference 19,20. (page 9, line167, 176)

The Authors stated “The major hypothesis on the etiology of AD is the amyloid cascade hypothesis”. Well, there are many (and very current) publications (review, in vitro, and in vivostudies) giving more relevance to the Aβ oligomer hypothesis (PMID: 29843241), which show how the toxicity/neurodegeneration observed in AD mainly depends on the formation of oligomeric species (PMID: 27804051; PMID: 30497524; PMID: 31293421). This “change of direction” should be emphasized in the Discussion section. Additionally, in certain conditions (e.g. sub-toxic levels or monomeric form) Aβ peptide could be protective for neurons by inducing the release of neurotrophic factors (PMID: 29094448; PMID: 30463298).

Reply: Thank you for your comments. We’ve revised as suggested. (page10, line186- 189)

The sentence “Whether the formations of senile plaques and NFTs are the causes of AD, or simply occur as a reactive response, remains unknown, although increasing evidence suggests that they may be induced by, and reflect, infection of the central nervous system” should be reformulated. Please define “NFT” before using it as an acronym.

Reply: Thank you for your comments. We’ve revised as suggested. (page 9, line 178- 181)

A reference should be added at the end of “Pathogenesis of leptospirosis also involves endothelial damage and vascular hypo-perfusion” and of “Leptospires frequently enter the body through abraded skin and are distributed into the bloodstream without the initial inflammatory response, followed by an immune phase characterized by antibody production”.

Reply: Thank you for your comments. We’ve added reference 36 (page 9, line 198, 194)

I could not find any connection with the Tables or Figures (except for Table 1). The Authors should refer to a specific table or figure (when it is pertinent) during their discussion, it will help the readers to better follow the “speech flow”.

A reference should be added at the end of “This finding revealed that leptospirosis might share at least some risk factors involved in the pathogenesis of depression, such as endothelium dysfunction”.

Reply: Thank you for your comments. It’s a mistake we mentioned about depression. It should be “ dementia”. We’ve revised it (page 10, line 208)

The meaning of the sentence “This kind of chronic exposure to leptospirosis maybe the pathogenesis of developing chronic endothelial damage, inflammatory processes, and the subsequent dementia” is not clear.

Reply: We tried to explain why an acute and curable infectious disease such as leptospirosis can lead to subsequent chronic disease such as dementia. It is not reasonable a person can develop dementia just due to one attack of leptospirosis. However, as we know, leptospirosis is an occupational and recreational disease. Persons who engage in occupational or recreational behavior that brings them into contact with Leptospira-infested habitats, may suffer from leptospirosis frequently with or without clear diagnosis. This kind of repeat and chronic exposure to leptospirosis maybe the pathogenesis of developing chronic endothelial damage, inflammatory processes, and the subsequent dementia.

The authors should tone down some of their statements, perhaps by deleting or reformulating “The association of dementia with leptospirosis and C. pneumoniae infection may also suggest that the vascular hypothesis may delineate the real pathogenesis of AD”. Additionally, why do they start talking about C. pneumoniae infection at some point? They have dedicated an entire paragraph to it (too much). It is distracting.

Reply: Thank you for your comments. We’ve deleted the all paragraph about C. pneumoniae.

Based on the proposed title “Risk of dementia in patients with leptospirosis: A nationwide cohort analysis”, the Discussion section related to AD pathology is unnecessarily long.

 Reply: Thank you for your comments. We’ve deleted the contents about the association between dementia and Chlamydia and reduced the contents about the association between dementia and other infection (HCV, HSV, etc…).

CONCLUSION

The conclusions are, at least in part, supported by Authors’ results. As highlighted by the Authors (I really like that they did this, almost nobody does it), “The present study had several limitations that must be considered”. As previously mentioned, if the Authors tone down some of their statements the present study could represent a good “pilot” for deeper future investigations.

 Minors:

Leptospira, Chlamydia pneumoniae, etc. should be “Italic”

Reply: Thank you for your comments. We’ve revised as suggested. (page4, line29) (page5, line74) (page11, line212) (page12, line255)

It not clear why some parts of the references are in bold. The Authors should be consistent with the references format.

Reply: Thank you for your comments. We’ve revised it.

Reviewer 2 Report

This paper presents important research, relatable globally. Overall, it is well written and employs a good research method to adequately satisfy the chosen question. Some more specific comments are below.

Line 50: There are more than 250 known pathogenic serovars of lepto Line 50: Rather than saying that lepto is known to be problematic in the tropics and developing countries , it would be worth specifying that it is a global problem - it is more prevalent in these areas but lepto occurs all over the world through occupational hazards, recreational hazards, waste hazards and contact with animals among other things. A little better information here would make it more relevant.  Line 124: Instead of saying "typical" male occupations, I would suggest saying "male dominated occupations".  Line 161: Reference?  Line 165: I would remove the words "As we know". Also is there a better word than "most?" This is not very specific.  Line 170: The word should be chose, not choose Line 178: What are NFTs? Please use the full word the first time. 

I feel as though the first few paragraphs on the discussion are a bit disjointed and dont link very well. It might need another sentence or two to create that link. 

The limitation are well described. The conclusion would benefit from the another sentence explaninig why this information is important and who it will benefit. 

Author Response

Reviewer #2:

This paper presents important research, relatable globally. Overall, it is well written and employs a good research method to adequately satisfy the chosen question. Some more specific comments are below.

Line 50: There are more than 250 known pathogenic serovars of lepto

Reply: Thank you for your comments. We’ve revised as suggested. (page4, line30)

Line 50: Rather than saying that lepto is known to be problematic in the tropics and developing countries , it would be worth specifying that it is a global problem - it is more prevalent in these areas but lepto occurs all over the world through occupational hazards, recreational hazards, waste hazards and contact with animals among other things. A little better information here would make it more relevant. 

Reply: Thank you for your comments. We’ve revised as suggested. (page4, line30-33)

Line 124: Instead of saying "typical" male occupations, I would suggest saying "male dominated occupations". 

Reply: Thank you for your comments. We’ve moved it to Discussion and revised as suggested. (page8, line155)

Line 161: Reference? 

Reply: Thank you for your comments. We’ve added reference 19 (page 9, line 167)

Line 165: I would remove the words "As we know". Also is there a better word than "most?" This is not very specific. 

Reply: Thank you for your comments. We’ve revised as suggested. (page 9, line162-163)

Line 170: The word should be chose, not choose

Reply: Thank you for your comments. We’ve revised as suggested. (page 9, line169)

Line 178: What are NFTs? Please use the full word the first time. 

Reply: Thank you for your comments. We’ve revised as suggested. (page 9, line179)

I feel as though the first few paragraphs on the discussion are a bit disjointed and dont link very well. It might need another sentence or two to create that link. 

Reply: Thank you for your comments. We’ve revised as suggested. (page 9, line157-158)

The limitation are well described. The conclusion would benefit from the another sentence explaninig why this information is important and who it will benefit. 

Reply: Thank you for your comments. We’ve revised as suggested. (page10, line261-263)

Round 2

Reviewer 1 Report

The manuscript was strongly improved by the Authors.